# Reproducibility of "Multi-scale Interactive Network for Salient Object Detection" for ML Reproducibility Challenge 2020

## 1    Reproducibility Summary

**Scope of Reproducibility**

The main claim of the original paper that our team tested was that their Multi-scale Interactive Network for Salient Object Detection model outperforms existing state-of-the-art SOD methods on the 5 mentioned datasets.

**Methodology**

Our team used the provided GitHub source from the authors, after downloading and installing all necessary dependencies. We first tested the model on the 5 given datasets in the paper then on 3 others by training on the same DUTS/Train dataset as mentioned in the paper. Since the provided code only computed three of the six measurement statistics reported on, a separate Saliency Evaluation Toolbox was utilized to compute the remaining measurements. While training the MINet model on our 3 datasets, we noticed some limitations. One being the results of all three F-measure statistics (Max-F, Mean-F and the Weighted-F) on both the SOC and THUR15K dataset were meaningless due to the datasets containing images with no salient object, so we attempted to improve upon the F-Measures. A related issue was that the THUR15K dataset could not be run through the MINet model because it does not include a mask for images with no salient object, so we wrote a script to generate a black mask. We then tested the model by training on the SOC training set and then on the combined DUTS and SOC training sets.

**Results**

After testing the results presented using the five datasets provided, we found good results that were close to the ones presented in the original paper. After checking the performance of the model on 3 additional datasets, we decided to modify the F-measures statistics as they were meaningless on the SOC dataset and recompute the measurements which gave us improved results. After training on the SOC set which contained images where the ground truth is a pure black mask, there was improvement on the SOC validation set and THUR15K. Training on the combined SOC and DUTS training sets saw overall good performance on all datasets except the THUR15K dataset, where the model is expected to not only identify salient objects, but if it is the object of interest.

**What was easy**

Since the author's made their Python source code publicly available on GitHub, testing the model was fairly quick. The code available was thorough and worked well. Additionally, as opposed to the training, the testing did not take much time and the measurement results were fairly good.

**What was difficult**

The biggest difficulty our team faced was the long training time and access to GPUs. For instance, training the MINet-VGG16 Model on the DUTS/Train dataset as in the paper took 63.5 hours. Lastly, the THUR15K dataset needed to be modified before we could test on it.

**Communication with original authors**

Our team did not communicate with the authors at all, except to use their publicly available source code.

# 1  Introduction

Salient object detection (SOD) methods have been of great use in recent years. This method distinguishes the most visually obvious features and region in an image. The applications of SOD include visual tracking, image retrieval, no-reference synthetic image quality assessment and so on [10] [13].

Despite the large progress that the SOD method reached, two challenges remain: how to extract more effective information from the data of scale variation and how to improve the spatial coherence of predictions in this situation.

# 2  Scope of reproducibility

To address the challenge of scale variance in SOD, this paper proposed the Multi-scale Interactive Network (MINet) which equipped with aggregate interaction modules (AIMs) and self-interaction modules (SIMs) [13]. AIMs exploit multi-level features and avoid the interference caused by resolution differences. SIMs obtain scale-specific information from the extracted features.

Besides, the paper introduced the consistency-enhanced loss (CEL) as the loss function. The CEL is insensitive to the scale of objects, handles the issue of spatial coherence better, and is able to highlight salient regions without the need of additional parameters.

For validation, the author trained and tested the MINet on five widely-adopted SOD datasets and the output were compared, using six SOD evaluation measures, to the output of 23 state-of-the-art saliency detection methods. The results suggested that the MINet outperformed all existing SOD methods across almost all datasets.

The claim from the original paper that our team tested is:

- The MINet outperforms existing state-of-the-art SOD methods on the 5 given datasets

# 3  Methodology

Since the author's source code is publicly available at https://github.com/lartpang/MINet, it was used as the basis for the implementation and training of the MINet. In the author's paper, there were six measurement statistics reported for the results on each dataset. However, the Python code only contained functions to calculate three of the measurements. The author's created lartpang/SODEvalToolkit, which was based on ArcherFMY/sal_eval_toolbox [6], both of which are available on GitHub. This toolkit contained functions implemented in MATLAB to compute the remaining measurement statistics on their dataset. For the testing, we added a measure module to compute the six measures within the Python code automatically during testing, which was based on Mehrdad-Noori/Saliency-Evaluation-Toolbox [11] [12], available on GitHub. Besides the five datasets used by the author, we also train and tested the model on the MSRA [2], the THURK15K [3], and the validation set of the SOC dataset [4].

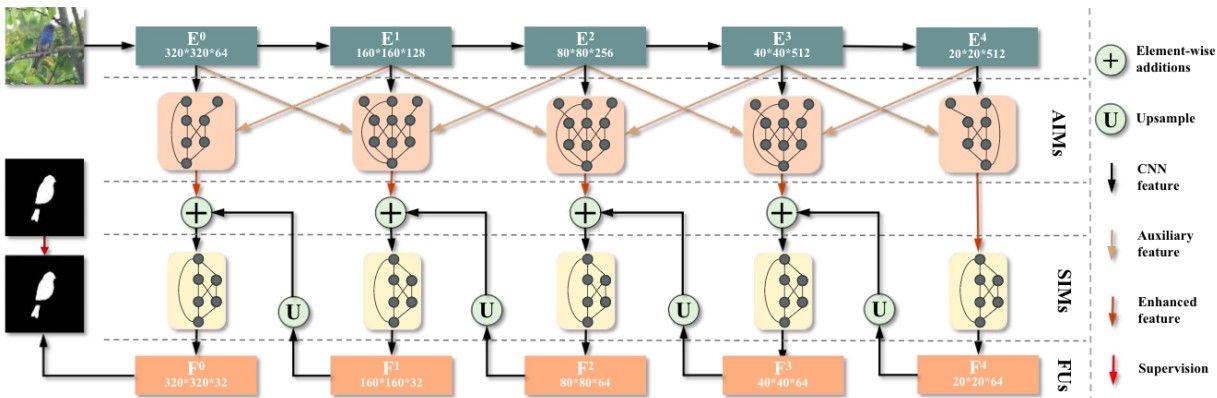

Figure 1: The overall framework of the proposed model [13]

### 3.1 Model descriptions

The MINet uses a pre-trained VGG-16 [15] network with a factor of 16 to extract multi-level features (or a ResNet-50 [7] with a factor of 32). Specifically, the last max-pooling layer of the VGG-16 is removed to maintain the details of the final convolution layer.

The extracted multi-level features are then fed into the Aggregate Interaction Modules (AIMs). AIMs enhance the features of one level by exploiting information from its adjacent levels. This differs from combining all level features, which produces information redundancy and noise interference. In the AIM, each node represents a combination of a single convolution layer, a batch normalization layer, and a ReLU layer. After the first layer, the branches are adjusted by element-wise addition with their two neighbors. Up-sampling or down-sampling is used to make the sizes of input from different resolutions match. Finally, the three branches are fused together, and the number of channels is reduced. This also employs a residual connection.

The output of AIMs is fed into Self-Interaction Modules (SIMs) as input to produce multi-scale representations from the intra-layer features. In the SIM, each node represents a combination of a convolution, normalization, and a ReLU layer. Up-sampling/down-sampling is used to ensure the same resolution as features from the next branch. Element-wise addition of the adjacent branch is then performed in the interactive layer. The Fusion Units (FU) fuses the features of the two paths from the SIM and the residual branch. The output of a SIM is then up sampled and added to the output of the AIM of the previous level (See Figure 1 taken from the author's paper [13]). The sum serves as an input of the SIM at the previous level.

For the loss function, the Consistency-Enhanced Loss is introduced to highlight the foreground region as smoothly as possible and to handle the sample imbalance issue. The total loss function is determined by summing the binary cross entropy loss and the consistency enhanced loss.

### 3.2 Datasets

For testing the proposed model, we used five popular existing benchmark datasets that this paper uses for code evaluation: DUTS[16], DUT-OMRON[18], ECSSD[17], HKU-IS[8], and PASCAL-S[9]. The DUTS contains 10,553 training and 5,019 test images, which is the largest salient object detection dataset. The DUT-OMRON contains 5,168 images of complex backgrounds and high content variety. The ECSSD is composed of 1,000 images with structurally complex natural contents. The HKU-IS contains 4,447 complex scenes that contain multiple disconnected objects with relatively diverse spatial distributions, and a similar background appearance makes it more difficult to distinguish. The PASCAL-S consists of 850 challenging images and 1,447 images for testing. Only the 850 challenging images from the PASCAL-S were used in our testing.

For testing whether the efficiency of the given code held in other datasets, we tested on three other datasets common for salient object detection area. We also tested our modifications on those three datasets. Firstly, the MSRA[2] dataset which originally provides salient object annotation in terms of bounding boxes. This dataset contains 10,000 images with consistent bounding box labeling and pixel-level saliency.

The second dataset our team chose was the THUR15K[3] dataset which contains about 3000 images for each of the 5 keywords: "butterfly", "coffee mug", "dog jump", "giraffe", and "plane", together comprising of 15,531 images. For each image, if there is a non-ambiguous object with correct content matching with the query keyword and most part of the object is visible, the object region is marked with a ground truth mask. Images that do not have non-ambiguous objects matching the query keyword do not have any associated ground truth mask. This dataset can also be used for evaluating co-segmentation methods.

Finally, the third dataset our team chose to test on was the SOC[4] Validation dataset, which overcomes design bias of other datasets which assume that each image contains at least one clearly outstanding salient object in low clutter. The SOC dataset has many images which do not contain any salient object and for which the ground truth is an empty black mask. The SOC dataset provides a total of 6,000 images with instance level salient object masks as well as attributes annotations. Of these images, 3600 are part of the Train set, 1200 part of the Validation set, and 1200 part of the Test set. However, we only tested on the Validation set as the Test set did not have complete ground truth data posted at the time of our testing.

### 3.3 Measurements

To validate the claim of the paper, we used the same six measures to evaluate the model. For all measures, the ground truth is first converted into a binary map by thresholding pixel intensities greater than 128 to 1 and below to 0. Note that this ignores the shades of grey in the ground truth mask of the PASCAL-S dataset.

**F-Measures:** To obtain a salient object's binary mask, we can threshold a saliency map at a specific threshold between 0 and 255 of the raw intensities or from 0 to 1 in 256 increments for the normalized map. From these, the true-positive (TP), true-negative (TN), false-positive (FP) and false-negative (FN) can all be calculated comparing the binary ground truth to the thresholded saliency map. The precision and recall are then calculated as:

$$\text{Precision} = \frac{TP}{TP + FP}$$

$$\text{Recall} = \frac{TP}{TP + FN}$$

The F-Measure $F_\beta$ is then calculated from the equation below. [1]

$$\text{F}_\beta\text{-measure} = \frac{(1 + \beta^2)\text{Precision} \cdot \text{Recall}}{\beta^2 \text{Precision} + \text{Recall}}$$

The $F_{max}$ denotes the maximal $F_\beta$ value when we vary the pixel threshold value from 0 to 1 in 256 increments. However, note that this is calculated by first averaging the precision and recall curves across all images in the dataset and then taking the max, not the other way around. This means that the maximum is still affected by every image in the dataset.

The $F_{avg}$ is calculated using an adaptive threshold that is twice the prediction's mean value for each image, and then these resulting F-measures are simply averaged over every image in the test set to produce the final $F_{avg}$ for the dataset.

The $F_\beta^\omega$ is calculated by replacing in the precision with weighted precision and recall with weighted recall. The weighted precision and recall are obtained by assigning different weights according to location and neighborhood information. The original F-Measure does not take into account if the pixels of the salient map sparsely cover the ground truth mask and therefore have good coverage, or if they are concentrated in one area with a segment of the object missing. The weighted F-measure will take this into account as well as weight errors (FP) in the mask differently depending on how far they are from the ground truth mask. The $F_\beta^\omega$ for each dataset is simply taken as the average of the $F_\beta^\omega$ for each image in the dataset.

**MAE:** The MAE is obtained by calculating the mean of the absolute value of the difference between the saliency map and the ground truth. Note that in the code, only the ground truth is first converted to a binary mask, while the saliency map is merely normalized between 0 and 1. Also note that in contrast to all other measurements, a lower number is better for the MAE. [14]

$$\text{MAE} = \frac{1}{W \times H} \sum_{x=1}^{W} \sum_{y=1}^{H} |M(x, y) - G(x, y)|$$

**E-Measure:** The E-Measure $E_m$ uses an enhanced alignment matrix to capture both pixel level and image statistics of a binary map when evaluating the similarity between the saliency map and the ground truth [5].

To capture both pixel and image level information, a bias matrix $\phi$ is defined as the distance between each pixel value of the binary foreground map and its global mean. And to quantify the bias matrix, an alignment matrix $\xi_{FM}$ is defined as the following equation where $\circ$ denotes the element-wise multiplication:

$$\xi_{FM} = \frac{2\varphi_{GT} \circ \varphi_{FM}}{\varphi_{GT} \circ \varphi_{GT} + \varphi_{FM} \circ \varphi_{FM}}$$

The enhanced alignment matrix is obtained by applying a mapping function $f$ that suppresses decrease at value regions and strengthens increase at positive regions to the alignment matrix:

$$\Phi_{FM} = f(\xi_{FM})$$

where

$$f(x) = \frac{1}{4}(x + 1)^2$$

Thus, the E-Measure $E_m$ can be calculated as the following where $h$ and $w$ are the height and the width of the map, respectively:

$$E_m = \frac{1}{w \times h} \sum_{x=1}^{w} \sum_{y=1}^{h} \Phi_{FM}(x, y)$$

**S-Measure:** The S-Measure $S_m$ is obtained by simultaneously computing the region-aware and object-aware structural similarities between a saliency map and the ground truth. [4]

To compute the region-aware structural similarity, the saliency map and the ground truth is recursively divided into multiple sub-blocks and each sub-block is assigned a weight that is proportional to the ground truth foreground region the sub-block covers. The region-aware structural similarity $S_r$ can be obtained from the equation:

$$S_r = \sum_{k=1}^{K} w_k \times \text{ssim}(k)$$

where K denotes the number of sub-blocks and ssim denotes the structural similarity measure which can be computed as the product of luminance comparison, contrast comparison and structure comparison.

To compute the object-aware structural similarity, we first need to compute the foreground comparison and background comparison between the saliency map and the ground truth. The foreground comparison $O_{FG}$ can be obtained from the equation:

$$O_{FG} = \frac{2\overline{x}_{FG}}{(\overline{x}_{FG})^2 + 1 + 2\lambda \times \sigma_{x_{FG}}}$$

where $\overline{x}_{FG}$ and $sigma_{x_{FG}}$ denote the means and standard deviation, respectively, of probability values of foreground region of the saliency map and the ground truth. $\lambda$ is a constant to balance the the saliency map and the ground truth.

Similar, The background comparison $O_{BG}$ can be obtained from the equation:

$$O_{BG} = \frac{2\overline{x}_{BG}}{(\overline{x}_{BG})^2 + 1 + 2\lambda \times \sigma_{x_{BG}}}$$

Let $\mu$ be the ratio of foreground area to image area, the object-aware structural similarity $S_o$ can be computed as:

$$S_o = \mu \times O_{FG} + (1 - \mu) \times O_{BG}$$

Thus, the S-Measure $S_m$ can be obtained with the equation below where $\alpha$ is set to 0.5 in the implementation:

$$S_m = \alpha \times S_o + (1 - \alpha) \times S_r$$

### 3.4 Modified F-Measures

After running the author's code on the SOC validation dataset and the THUR15K datasets, it was clear that the F-measures were misleading and returning unexpectedly low values on these datasets. After some investigation, it appears this was because the F-measures were not well defined when the ground truth was a purely black mask, indicating that there was no salient object in the image. Both the SOC and THUR15K contain such images where the model is expected to recognize that there *is not* any salient object and as such produce a black saliency map.

When the ground truth is an "empty" or purely black mask, then the precision and recall curves that the F-measure depends on are ill-defined. In this case, both the True Positive and and the False Negative were always 0 because there are no pixels of interest in the ground truth mask. This results in the precision being 0 and the recall being 0/0 and ill defined. The author's MINet code handles this by simply setting both the precision and recall to 0, which then affects the resulting F-Measure statistics, as they are averaged across all images in the dataset. Therefore, the results of all three F-measure statistics (Max-F, Mean-F and the Weighted-F) on both the SOC and THUR15K dataset are substantially lowered by every instance where the ground truth is a black mask, resulting in these measures being meaningless, and misleading to the uninformed user.

In an attempt to prevent the F-Measures from producing meaningless results for datasets that contain images without salient objects, our team attempted to extend the F-Measure to use an alternative measurement for said images. In the case when the ground truth is a black mask and has no pixels of interest, we propose a modification to the F-measure:

$$\text{Modified-F}_\beta = \begin{cases} F_\beta, & \sum_{x=1}^{W}\sum_{y=1}^{H} G(x,y) \neq 0 \\ (1-\text{MAE})^n, & \sum_{x=1}^{W}\sum_{y=1}^{H} G(x,y) = 0 \end{cases}$$

Where $n$ is a parameter to control the weighting of errors in the saliency map compared to the black mask. We found that using a value of $n = 2$ resulted in reasonable results where the F-measure is no longer quite so misleading in the datasets which contain images with no salient object.

For the $F_{max}$, a modified approach was needed since this measurement is calculated by first averaging the precision and recall curves of all images in the dataset and then computing the F-measures. In this case, we modify the Precision and Recall to be calculated in reference to producing a pure black mask as below:

$$\text{Precision} = \left( \frac{1}{W \times H} \sum_{x=1}^{W}\sum_{y=1}^{H} S(x,y) \right)^n$$

$$\text{Recall} = \frac{1}{W \times H} \sum_{x=1}^{W}\sum_{y=1}^{H} S(x,y)$$

Where $S(x,y)$ is a binary mask first calculated by thresholding in an inverse manner where $M(x,y) \leq \gamma$, where $\gamma$ is a threshold from 0 to 1 in 256 increments. We also find $n = 2$ gives decent results. As such, the recall is calculated based on recalling a pure black image where the Recall = 1 when the saliency map is purely black. The precision and recall are then averaged over all images, where they are calculated the same as originally when there is a salient object in the ground truth and as described here when there is not. The $F_{max}$ is then computed in the same manner as before. Although these are not "true" precision and recall definitions, they produce more meaningful results then simply setting them to 0.

### 3.5 Hyperparameters

We double-checked the hyperparameters of the source code provided and decided to leave most of them as the default as they were the same as reported in the author's paper. The main hyperparameter settings are summarized in Table 1 and these were not changed between any of the test runs.

Table 1: Main hyperparameters used when training the MINet

| Hyperparameter | Value | Comments in Author's Code |
|---|---|---|
| Epoch number | 50 | None |
| Learning rate | 0.001 | None |
| Tensorboard update | 50 | if the value is >0, will use tensorboard |
| Print Frequency | 50 | if >0, the code will save the information into a file |
| Size list | None | Not using multi-scale training |
| Reduction | "mean" | How to handle reduction,'mean' or 'sum' |
| Weight decay | 5e-4 | When fine-tuning, set to 0.0001 |
| Momentum | 0.9 | None |
| Learning rate type | 'poly' | None |
| Warm-up epoch | 1 | when set to 1, it means there is no warm-up |
| Learning rate decay | 0.9 | for the poly type |
| Batch size | 4 | Keep the same batch size for all datasets |
| Number worker | 4 | If this value is too big, it will impact the speed of data reading |
| Input size | 320 | None |

## 3.6 Experimental setup

Since the author's source code is publicly available, it was used as the basis for the implementation and training of the MINet. However, as previously mentioned, a separate toolbox was needed in order to calculate the six measurement statistics on the resulting images. We downloaded and integrated the Saliency Evaluation Toolbox from https://github.com/Mehrdad-Noori/Saliency-Evaluation-Toolbox into the author's code to compute all the measurements statistics in Python when testing on the test sets. At a minimum, this involved modifying their solver.py file which contained the test function as well as the recorder.py file which outputted the results to an Excel spreadsheet.

Also, in order to run their code, some Python packages needed to be installed first. These are listed below along with the versions we installed (not necessarily the same version as the authors).

- Python = 3.7
- PyTorch = 1.7.0
- torchvision = 0.8.1
- Pillow = 8.0.1
- tqdm = 4.52.0
- prefetch-generator = 1.0.1
- tensorboard = 2.4.0
- openpyxl = 3.0.5
- scipy = 1.5.2

Our fork of the author's code is publicly available on Github at https://github.com/Farzanehkaji/MINet.

Since the MINet model was tested using two backbone models (VGG-16 and ResNet-50) in the paper, we started by verifying the results on each. The MINet was trained on the same DUTS/Train dataset as in the paper using first the VGG-16 and then the ResNet-50 backbone, using the same hyperparameters described in the author's paper. After training, the models were tested on the 8 datasets mentioned previously, of which 5 coincide with those tested in the author's paper. Testing included both using the model to make prediction saliency maps for each test image, as well as computing the measurements statistics on the resulting maps. We also computed the modified F-measure statistics we propose to see their result.

We then tested training the MINet model with the ResNet-50 backbone on different training sets that contain images with no salient object and where the associated ground truth is simply a pure black mask. These tests included training on just the SOC/Train set and then the DUTS/Train and SOC/Train sets combined for a total of 14,153 training images to see if performance on the SOC/Validation set and the THUR15K set increased.

As the network is quite large, it was essential that the code be run on a computer with a NVIDIA CUDA-enabled GPU. See below for the computer specifications and training times.

## 3.7 Computational requirements

Table 2 represents a summary of the computational time required for training and testing the model on all 8 data-sets. The test time is the total time for the testing the model across all 8 data-sets and to compute all reported measurement statistics, both the original six measurements and our modified versions of the F-measures discussed later.

Table 2: Training and testing time for each model tested on 8 data-sets with 40,215 images in total

| Run | Base Model | Training Set | Training Set Size | Training Time (hh:mm) | Test Time[1] (hh:mm) |
|---|---|---|---|---|---|
| 1 | VGG-16 | DUTS/Train | 10,553 | 63:29 | 3:32 |
| 2 | ResNet-50 | DUTS/Train | 10,553 | 32:32 | 3:34 |
| 3 | ResNet-50 | SOC/Train | 3,600 | 10:50 | 3:39 |
| 4 | ResNet-50 | DUTS/Train + SOC/Train | 14,153 | 17:50 | 3:07 |

---

[1]The final test time is estimated as some output measurement statistics were computed during testing and further measurements were added and computed on the already made output predictions later.

See a summary of the computer specifications in Table 3. Note that the first three runs were done on a personal machine whereas the final run, which was trained on the combined DUTS and SOC training sets, was trained and tested using an Amazon AWS EC2 p3.2xlarge instance.

Table 3: Computer Specifications

| Runs | OS | GPU | GPU Memory | CPU | RAM |
|------|------|------|------|------|------|
| 1 - 3 | Windows 10 | NVIDIA GTX 1070 | 8 GB | Intel i7-7700k | 32 GB |
| 4 | Ubuntu 18.04 | NVIDIA V100 | 16 GB | 8 vCPUs | 61 GB |

## 4 Results

The results were divided into 4 separate sections where we discuss the measurement results after each modification that was made. Firstly, we tested using the MINet with both the VGG-16 and ResNet-50 backbones trained on the DUTS/Train dataset using the same five test datasets in the paper as well as an additional three datasets the team chose for testing. Secondly, after creating the modified F-measures due to the poor results from the SOC dataset, we computed the modified F-Measures on the output predictions to check if the measurements were more meaningful. The third section discusses the training results on the SOC Training dataset to check if the results improved on the SOC validation set and the THUR15K dataset after training on a dataset that contains images where there is no salient object and as such have an associated pure black mask. Finally, the last section discusses the results of training the MINet-Res50 on the combined SOC and DUTS training sets.

### 4.1 Results from training on DUTS/Train and testing on 8 datasets

We started our experiments by testing the unchanged MINet model with a VGG-16 backbone trained on the DUTS trained dataset to compare with the results given by the paper and test the performance of the unchanged model. We first tested the MINet model on the 5 datasets similar to the paper's experiment: DUTS, DUT-OMRON, HKU-IS, ECSSD, and PASCAL-S. Then we tested on the three additional datasets our team chose: MSRA10K, SOC, and THUR15K. As can be seen in Table 4 below, our testing values on the first 5 datasets were very close to those documented in the original paper, thus demonstrating that their results are reproducible. The S-Measure values are the only measurements that are slightly lower, though this could be due to slight differences in how they were calculated as we used a different toolbox for the calculation as the author's, and all the other measurements are similar. As for the other three datasets, the results were similarly good on the MSRA10K set, but performance was worse on the SOC validation set and the THUR15K set. Refer to Figures 2 and 3 for sample saliency masks output images of this test in the 3rd column.

Table 4: Run 1 - Testing MINet with VGG-16 backbone trained on DUTS/Train

| | Data-Set | $F_{max}$ | $F_{avg}$ | $F_\beta^\omega$ | MAE | $E_m$ | $S_m$ |
|------|------|------|------|------|------|------|------|
| Paper's Results | DUTS | 0.877 | 0.823 | 0.813 | 0.039 | 0.912 | 0.875 |
| | DUT-OMRON | 0.794 | 0.741 | 0.719 | 0.057 | 0.864 | 0.822 |
| | HKU-IS | 0.932 | 0.906 | 0.892 | 0.030 | 0.955 | 0.914 |
| | ECSSD | 0.943 | 0.922 | 0.905 | 0.036 | 0.947 | 0.919 |
| | PASCAL-S | 0.882 | 0.843 | 0.820 | 0.065 | 0.898 | 0.855 |
| Our Testing | DUTS | 0.878 | 0.821 | 0.814 | 0.039 | 0.912 | 0.847 |
| | DUT-OMRON | 0.793 | 0.737 | 0.716 | 0.057 | 0.862 | 0.792 |
| | HKU-IS | 0.932 | 0.904 | 0.890 | 0.031 | 0.955 | 0.890 |
| | ECSSD | 0.943 | 0.920 | 0.904 | 0.036 | 0.948 | 0.896 |
| | PASCAL-S | 0.869 | 0.829 | 0.810 | 0.063 | 0.897 | 0.838 |
| Other Data-Sets | MSRA10K | 0.916 | 0.894 | 0.869 | 0.049 | 0.938 | 0.879 |
| | SOC | 0.385 | 0.360 | 0.344 | 0.092 | 0.834 | 0.835 |
| | THUR15K | 0.324 | 0.300 | 0.296 | 0.187 | 0.792 | 0.765 |

Table 5 presents the results of the testing with ResNet-50 with the backbone trained on the DUTS dataset. Similar to the previously discussed results, we notice that our testing values are very close to the ones presented by the original authors. Our tests showed slightly lower measurements in comparison to the paper's results when looking the F-Measures on

the PASCAL-S for example, but then our tests resulted in better performance on the same dataset when using other statistics such as the MAE and E-Measure. As for our additional datasets' measurement values, these were similar to the ones from the previous run in Table 5 further showcasing the good performance of the given MINnet model on the MSRA10K set, and lower performance on the SOC validation set and THRU15K sets. Both the VGG16 and Resnet50 models performed very well on the MSRA10K getting similar results to the five datasets already given. Refer to Figures 2 and 3 for sample saliency masks output images of this test in the 4th column.

Table 5: Run 2 - Testing MINet with ResNet-50 backbone trained on DUTS/Train

|  | Data-Set | $F_{max}$ | $F_{avg}$ | $F_\beta^\omega$ | MAE | $E_m$ | $S_m$ |
|---|---|---|---|---|---|---|---|
| Paper's Results | DUTS | 0.884 | 0.828 | 0.825 | 0.037 | 0.917 | 0.884 |
|  | DUT-OMRON | 0.810 | 0.756 | 0.738 | 0.055 | 0.873 | 0.833 |
|  | HKU-IS | 0.935 | 0.908 | 0.899 | 0.028 | 0.961 | 0.920 |
|  | ECSSD | 0.947 | 0.924 | 0.911 | 0.033 | 0.953 | 0.925 |
|  | PASCAL-S | 0.882 | 0.842 | 0.821 | 0.064 | 0.899 | 0.857 |
| Our Testing | DUTS | 0.887 | 0.837 | 0.829 | 0.036 | 0.922 | 0.860 |
|  | DUT-OMRON | 0.805 | 0.756 | 0.736 | 0.053 | 0.874 | 0.811 |
|  | HKU-IS | 0.935 | 0.910 | 0.896 | 0.029 | 0.959 | 0.894 |
|  | ECSSD | 0.947 | 0.927 | 0.910 | 0.034 | 0.953 | 0.902 |
|  | PASCAL-S | 0.873 | 0.837 | 0.817 | 0.061 | 0.903 | 0.845 |
| Other Data-Sets | MSRA10K | 0.920 | 0.900 | 0.877 | 0.047 | 0.941 | 0.887 |
|  | SOC | 0.391 | 0.363 | 0.347 | 0.106 | 0.806 | 0.824 |
|  | THUR15K | 0.324 | 0.302 | 0.298 | 0.186 | 0.793 | 0.767 |

## 4.2 Results using modified F-Measures

Running the given code for the SOC validation set of 1200 images gave us poor results. One of the reasons behind that was that the F-Measures were not well defined with the ground truth image being pure black mask since there is no salient object. In this case, both the True Positive and and the False Negative were always 0 because there are no pixels of interest in the ground truth mask. This results in the recall being 0/0 and ill defined. The author's MINet code handles this by simply setting both the precision and recall to 0, which then affects the resulting F-Measure statistics, as they are averaged across all images in the dataset. Therefore, the results of all three F-measure statistics (Max-F, Mean-F and the Weighted-F) on both the SOC and THUR15K dataset were meaningless. From the MAE measures, which are still meaningful when the ground truth is a pure black mask, the model didn't perform well on the SOC validation set; for many of the images where there were no salient objects, the model produces somewhat unpredictable results and will sometimes produce a mask that covers a large portion of the image. Refer to Figure 3 to see examples from the SOC Validation set and the THUR15K set where the ground truth image is a pure black mask.

Table 6: Run 1 - Testing MINet with VGG-16 backbone trained on DUTS/Train with modified F-Measures

|  | Dataset | $F_{max}$ | $F_{avg}$ | $F_\beta^\omega$ | MAE | $E_m$ | $S_m$ | $F_{max}{'}$ | $F_{avg}{'}$ | $F_\beta^{\omega}{'}$ |
|---|---|---|---|---|---|---|---|---|---|---|
| Datasets without empty masks | DUTS | 0.878 | 0.821 | 0.814 | 0.039 | 0.912 | 0.847 | 0.878 | 0.821 | 0.814 |
|  | DUT-OMRON | 0.793 | 0.737 | 0.716 | 0.057 | 0.862 | 0.792 | 0.793 | 0.737 | 0.716 |
|  | HKU-IS | 0.932 | 0.904 | 0.890 | 0.031 | 0.955 | 0.890 | 0.932 | 0.904 | 0.890 |
|  | ECSSD | 0.943 | 0.920 | 0.904 | 0.036 | 0.948 | 0.896 | 0.943 | 0.920 | 0.904 |
|  | PASCAL-S | 0.869 | 0.829 | 0.810 | 0.063 | 0.897 | 0.838 | 0.875 | 0.836 | 0.816 |
|  | MSRA10K | 0.916 | 0.894 | 0.869 | 0.049 | 0.938 | 0.879 | 0.916 | 0.894 | 0.869 |
| Datasets with empty masks | SOC | 0.385 | 0.360 | 0.344 | 0.092 | 0.834 | 0.835 | 0.818 | 0.788 | 0.771 |
|  | THUR15K | 0.324 | 0.300 | 0.296 | 0.187 | 0.792 | 0.765 | 0.715 | 0.642 | 0.638 |

To allow for the instances where the ground truth is a pure black mask, we compute the modified F-measures as previously discussed. As can be seen in Table 6, there is a clear difference in the F-measures of the datasets that didn't contain empty masks (DUTS, DUT-OMRON, HKU-IS, ECSSD, PASCAL-S, and MSRA10K) and the ones that did (SOC and THUR15K). The F-measures of the SOC and THUR15K datasets were markedly lower than other datasets. After computing the modified F-measures, we can see that the datasets SOC and THUR15K standout in their Modified

F-measures $F_{max}'$, $F_{avg}'$, and $F_\beta^{\omega}{}'$ values being very different from their respective $F_{max}$, $F_{avg}$, and $F_\beta^{\omega}$ values as opposed to the other datasets where the Modified-F-Measures and original F-Measures are the same. When the datasets do not contain any instances where the ground truth saliency map is empty, there is no difference in the modified F-measures vs the original, as desired. However, in cases where the dataset does contain instances where the ground truth is a pure black image, the modified F-Measures provide a more meaningful statistic which is correspondingly inversely proportional to the MAE. Note that the PASCAL-S dataset is an exception where it does not contain any instances where the ground truth is empty, but it does contain shades of grey maps on the ground truths which may have been thresholded to black in some instances.

Table 7: Run 2 - Testing MINet with ResNet-50 backbone trained on DUTS/Train with modified F-Measures

| | Dataset | $F_{max}$ | $F_{avg}$ | $F_\beta^{\omega}$ | MAE | $E_m$ | $S_m$ | $F_{max}'$ | $F_{avg}'$ | $F_\beta^{\omega}{}'$ |
|---|---|---|---|---|---|---|---|---|---|---|
| Datasets without empty masks | DUTS | 0.887 | 0.837 | 0.829 | 0.036 | 0.922 | 0.860 | 0.887 | 0.837 | 0.829 |
| | DUT-OMRON | 0.805 | 0.756 | 0.736 | 0.053 | 0.874 | 0.811 | 0.805 | 0.756 | 0.736 |
| | HKU-IS | 0.935 | 0.910 | 0.896 | 0.029 | 0.959 | 0.894 | 0.935 | 0.910 | 0.896 |
| | ECSSD | 0.947 | 0.927 | 0.910 | 0.034 | 0.953 | 0.902 | 0.947 | 0.927 | 0.910 |
| | PASCAL-S | 0.873 | 0.837 | 0.817 | 0.061 | 0.903 | 0.845 | 0.878 | 0.843 | 0.823 |
| | MSRA10K | 0.920 | 0.900 | 0.877 | 0.047 | 0.941 | 0.887 | 0.920 | 0.900 | 0.877 |
| Datasets with empty masks | SOC | 0.391 | 0.363 | 0.347 | 0.106 | 0.806 | 0.824 | 0.810 | 0.770 | 0.755 |
| | THUR15K | 0.324 | 0.302 | 0.298 | 0.186 | 0.793 | 0.767 | 0.715 | 0.645 | 0.641 |

## 4.3 Results from training on SOC/Train dataset

In this section, we look at the results of training the MINet with the ResNet-50 backend using the SOC training dataset, which only contains 3600 images compared to the 10,551 in the DUTS/Train. The SOC set however contains instances where the ground truth is a pure black mask, and as such should help to train the model to identify when there is no salient object. The resulting measurements from testing do indeed show an improvement on the SOC validation set as the code was able to return black image, no salient object detected, for certain cases. Note that the THUR15K dataset also has some confusing images that did not contain the object of one of the five classes, but instead had something that looked similar. In such cases, the similar looking object would still be detected as a salient object by this model, because it was not trained to only detect objects of those five classes. As can be seen in Table 8 below, there are improvement in the measurement values for both the SOC and THURK15K datasets in comparison with the measurements in Table 7. However, performance was still fairly poor on the THUR15K dataset. Refer to Figures 2 and 3 for sample saliency masks output images of this test in the 5th column.

Note however, that after training on the SOC validation set, performance was reduced on the first six datasets in the table which did not contain any instances where the ground truth was empty. This could be associated with the much smaller size of the SOC training set compared to the DUTS training set.

Table 8: Run 3 - Testing MINet-Res50 trained on SOC/Train with modified F-Measures

| | Dataset | $F_{max}$ | $F_{avg}$ | $F_\beta^{\omega}$ | MAE | $E_m$ | $S_m$ | $F_{max}'$ | $F_{avg}'$ | $F_\beta^{\omega}{}'$ |
|---|---|---|---|---|---|---|---|---|---|---|
| Datasets without empty masks | DUTS | 0.814 | 0.769 | 0.691 | 0.058 | 0.855 | 0.789 | 0.814 | 0.769 | 0.691 |
| | DUT-OMRON | 0.701 | 0.645 | 0.548 | 0.088 | 0.763 | 0.725 | 0.701 | 0.645 | 0.548 |
| | HKU-IS | 0.877 | 0.848 | 0.779 | 0.057 | 0.892 | 0.826 | 0.877 | 0.848 | 0.779 |
| | ECSSD | 0.869 | 0.835 | 0.761 | 0.078 | 0.859 | 0.818 | 0.869 | 0.835 | 0.761 |
| | PASCAL-S | 0.855 | 0.820 | 0.780 | 0.071 | 0.877 | 0.822 | 0.862 | 0.827 | 0.787 |
| | MSRA10K | 0.819 | 0.780 | 0.683 | 0.096 | 0.820 | 0.783 | 0.819 | 0.780 | 0.683 |
| Datasets with empty masks | SOC | 0.399 | 0.378 | 0.356 | 0.041 | 0.572 | 0.888 | 0.898 | 0.877 | 0.855 |
| | THUR15K | 0.307 | 0.289 | 0.274 | 0.142 | 0.807 | 0.799 | 0.771 | 0.704 | 0.688 |

## 4.4 Results from training on Combined SOC and DUTS Training datasets

Finally, we tested the MINet-Res50 model trained on combined SOC and DUTS Train with the modified F-measures on all the datasets. Again there was a clear distinction in the results of the datasets that didn't contain empty masks and the one that did, as shown in Table 9. In this case, performance on the first six datasets which did not contain any empty

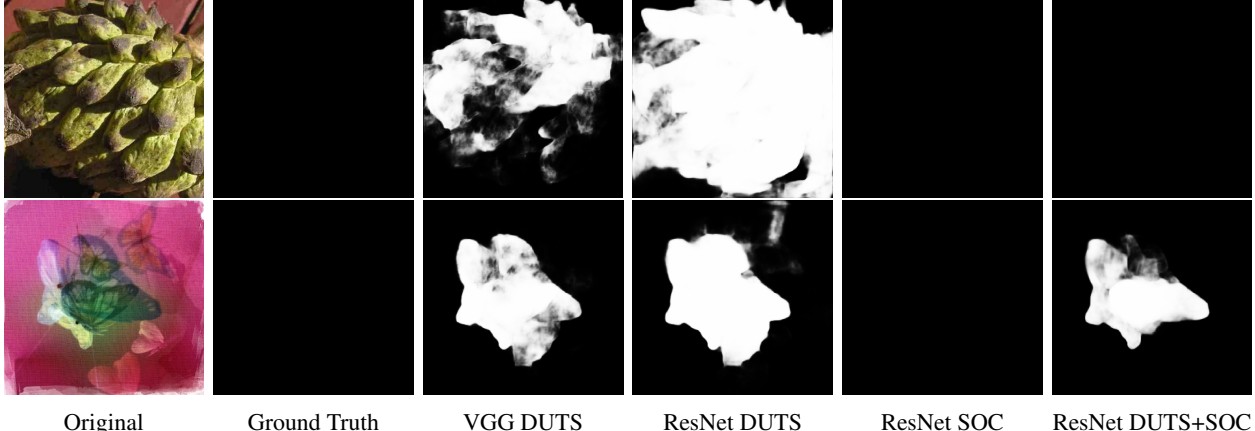

| Original | Ground Truth | VGG DUTS | ResNet DUTS | ResNet SOC | ResNet DUTS+SOC |

Figure 2: Test images for MINet trained on DUTS and SOC datasets

| Original | Ground Truth | VGG DUTS | ResNet DUTS | ResNet SOC | ResNet DUTS+SOC |

Figure 3: Test images with black ground truth for MINet trained on DUTS and SOC datasets

ground truth masks was improved from those shown in Table 8 where the training set was smaller. The performance on both the SOC validation set and the THRU15K was also improved compared to Table 7 where the model was trained on the DUTS/Train alone. This shows that the MINet model is fairly robust and can be trained for a slightly different intent to both identify when there is no salient object, and if there is, to generate a saliency map.

However, again, the overall performance on the THUR15K dataset is still relatively weak due to the confusing images this dataset contains, where there is an object, but it is not of the class denoted by the folder name and rather only looks similar. From the dataset description, the THUR15K dataset is meant to evaluate shape based image retrieval performance, so a method that is intended to look for salient objects which standout may not perform well as it does not necessarily care about what object it is seeing. Refer to Figures 2 and 3 for sample saliency masks output images of this test in the last column.

Table 9: Run 4 - Testing MINet-Res50 trained on combined SOC and DUTS Train with modified F-Measures

|  | Dataset | $F_{max}$ | $F_{avg}$ | $F_\beta^\omega$ | MAE | $E_m$ | $S_m$ | $F_{max}{}'$ | $F_{avg}{}'$ | $F_\beta^{\omega}{}'$ |
|---|---|---|---|---|---|---|---|---|---|---|
| Datasets without empty masks | DUTS | 0.885 | 0.839 | 0.822 | 0.036 | 0.922 | 0.857 | 0.885 | 0.839 | 0.822 |
|  | DUT-OMRON | 0.804 | 0.761 | 0.732 | 0.052 | 0.872 | 0.812 | 0.804 | 0.761 | 0.732 |
|  | HKU-IS | 0.928 | 0.908 | 0.881 | 0.032 | 0.951 | 0.885 | 0.928 | 0.908 | 0.881 |
|  | ECSSD | 0.937 | 0.918 | 0.896 | 0.037 | 0.942 | 0.893 | 0.937 | 0.918 | 0.896 |
|  | PASCAL-S | 0.870 | 0.842 | 0.812 | 0.060 | 0.900 | 0.843 | 0.877 | 0.849 | 0.819 |
|  | MSRA10K | 0.909 | 0.888 | 0.857 | 0.052 | 0.928 | 0.877 | 0.909 | 0.888 | 0.857 |
| Datasets with empty masks | SOC | 0.409 | 0.382 | 0.367 | 0.041 | 0.549 | 0.892 | 0.903 | 0.877 | 0.862 |
|  | THUR15K | 0.321 | 0.301 | 0.295 | 0.172 | 0.787 | 0.780 | 0.717 | 0.667 | 0.661 |

## 5 Discussion

Given the number of time our team trained different models on many datasets, time was definitely the most challenging aspect of our process. However, we did manage to run all the experiments and make additional modifications that further strengthened the claims in the paper. The evidence we got from running both the original code and the same DUTS/Train dataset and training it on other datasets were able to further support the claims of the paper; the MINet performs well on the mentioned datasets and outperforms many existing state-of-the-art SOD methods.

### 5.1 What was easy

Since the author's source code is publicly available, testing the network architecture was made easier since it did not need to be coded from scratch. The code available was thorough and worked well. The included readme files also significantly helped with understanding the code and how to run it. Although it took some time to download and install some of requirements to run the code, we didn't face any issues training and testing the code on the MINet model on the 5 given datasets.

Additionally, the testing was fairly easy on all the datasets we worked with. Their code implementation allowed specifying any number of dataset paths to test on, as long as the datasets were organized into an "Image" folder of the original images, and a "Mask" folder containing the ground truth masks. As opposed to the training, the testing did not take as much time and the measurement results were always good.

### 5.2 What was difficult

For the THUR15K dataset, we were not able to run it through the MINet model unchanged because it does not include a mask for every image. Additionally, the THUR15K organizes the images into 5 different categories and numbers them from 1 to 3000 in each folder, for a total of 15531 images. So there were around 5 duplicates for each filename. As such, we had to first write a script which would go through every image in the dataset and if it did not contain a ground truth saliency map, create a pure black one for it. The script also renamed the files by adding the folder name to the filename so that there were no duplicates and all of the dataset images could be moved into one folder as required by the MINet.

One of the biggest challenges was that the paper did not discuss the total training and testing time for their model, so we did not know how long it would take going in. Many of the comments in the code were also not in English, so it took our team some time to understand the different steps of the code and compare it with the model described in their paper. However, our team was able to understand it without needing to contact the authors with questions. The training

and testing also took a some time, both before and after modifications. Training the VGG-16 Model on the five datasets in the paper took 63.5 hours. Testing the original 5 datasets reported on in the paper with the 3 measures included in the code (F-Max, F-Mean, and MAE) took 65 min. However, after adding the extra datasets and testing the 9 total measures (with the modified F-measures), testing time increased to over 3 hours on our machine.

## 5.3 Communication with original authors

Our team didn't communicate with the original authors. We didn't feel the need to ask the original authors any questions to understand their model or code as the information provided in the paper as well as the GitHub code and readme files were very thorough. We utilized their public repository on GitHub to test the the algorithms.

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
