# OpenReview forum: "Reproducibility of "Multi-scale Interactive Network for Salient Object Detection" for ML Reproducibility Challenge 2020"
_ML_Reproducibility_Challenge/2020 — Reject_

### Official Review · AnonReviewer1 · 2021-02-26
**Off the shelf code rerun with 3 additional datasets.**

**Rating:** 6
**Confidence:** 3

**Review:**

The report aims to reproduce the following paper: "Multi-scale Interactive Network for Salient Object Detection".

The report is based on re-running the existing code-base. The report adds additional 3 datasets.

On the positive side, the report is clearly written and easy to follow and contains enough details to understand the original paper.

On the negative side, the experimental evaluation does not add new hyperparameters to test limiting the scope of the report given that it just re-runs the original paper implementation. Moreover, I'm not convinced about the F-measure argument, that it is not suitable to handle properly "black images", since bot precision and recall should handle properly images without silent objects. Maybe the issue is in the implementation of the F-measure?

**Familiar With The Original Paper:**

I have not read the original paper

**Reproducibility Summary:**

Report has summary

---

### Official Review · AnonReviewer2 · 2021-02-27
**Evaluates perfomance on three additional datasets**

**Rating:** 6
**Confidence:** 3

**Review:**

Overview:
The authors reproduce the paper using officially released github repo for MINet. They tested the approach extensively by trying it out on 3 additional datasets with necessary dataset-specific metric fixes.

Positives:
1. The authors test the method on three additional datasets.

Negatives:
1. The authors should try to optimize hyperparameters on the reported datasets since the code is already available. I do understand that there might be time/resource constraints. (Minor)
2. The numbers on the additional datasets in itself do not reveal much unless compared with other approaches. It would be great to test EGNet or SCRN on the additional datasets (whichever is less effort) and include in the report. If the results are publicly available in a different paper, it would be best to just add it to the report for completeness.

**Familiar With The Original Paper:**

I have not read the original paper

**Reproducibility Summary:**

Report has summary

---

### Official Review · AnonReviewer3 · 2021-03-02
**Reasonable RC report**

**Rating:** 6
**Confidence:** 3

**Review:**


The scope of the report is stated in the paper, however, it does not break down the scope into detailed items. It would be better to have a more detailed scope definition.

The code is from the original authors. The authors of this paper made some efforts in trying the proposed approach in different datasets.

Not much communication with the original authors other than reuse the GitHub code from the original authors.

There is a hyperparameter described in the paper, but there is not much hyperparameters search in the report.

It is interesting to see the comparison between different computer specifications to run the experiments, though they are not motivated well. I think it would be better if the authors could find out which salient region detection approach works the best for the retrieval task. Specifically, can you tweak the hyperparameters of the proposed approach? Would that change the retrieval performance?

I recognize that the paper has done a bit more experiments to check the various versions of the proposed approach on different datasets. There is a detailed discussion.

The original paper is reproducible. However, according to this paper, it is better to provide details about the time spent on the training.

According to Table 2, it seems that the reports have attempted some backbone models. I am wondering why we skip efficient networks like MobileNet v2 here if we are looking for a bit more efficient way in training and testing.

It is a bit confusing which computer in table 3 is used in reporting the training time and testing time in table 2. Also, it would be better if the two machine has the same CPUs or the same GPUs, so that we can have some control groups. Right now, there are too many differences in the two computers.

**Familiar With The Original Paper:**

I have read the original paper

**Reproducibility Summary:**

Report has summary

---

### Decision · Program_Chairs · 2021-03-31

**Decision:**

Reject

**Comment:**

Overall reviews and/or the paper content not good enough for the AC to recommend to the journal.